# Chromatographic Analysis of Bioactive Metabolites from a Traditional Food Combination of (Semi) Arid Regions—*Panchkuta*: Insights for Sustainable Functional Foods Development †

Tripti Joshi , P. R. Deepa and Pankaj Kumar Sharma *

Biochemistry and Enzyme Biotechnology Lab, Department of Biological Sciences, Birla Institute of Technology and Science, Pilani—Pilani Campus, Pilani 333031, Rajasthan, India; p20190441@pilani.bits-pilani.ac.in (T.J.)
* Correspondence: pankajsharma@pilani.bits-pilani.ac.in
† Presented at the 4th International Electronic Conference on Nutrients (IECN 2024), 16–18 October 2024; Available online: https://sciforum.net/event/IECN2024.

**Abstract:** Conventional agriculture may not meet the needs of the growing human population and sustainable development. These issues necessitate a shift towards traditional foods and underutilized desert plants, offering promising sustainable agricultural and food security alternatives. '*Panchkuta*', a blend of five plants (*Prosopis cineraria*, *Acacia senegal*, *Capparis decidua*, *Cordia dichotoma*, and *Mangifera indica*), is a traditional food combination from (semi) arid regions. In this study, the bioactive metabolites were identified through analytical techniques, including TLC and GC-MS analysis. Tapping these underexplored plants can help design and develop healthy functional foods and nutraceutical products.

**Keywords:** traditional food; underutilized desert plants; antioxidant; nutraceutical; functional food

## 1. Introduction

Plants serve as valuable resources for creating high-value products used across cosmetics, pharmaceuticals, fragrances, and food industries. Despite this potential, research indicates that <1% of plant species have been fully explored for their medicinal uses, and a significant number remain unexamined even in terms of their phytochemical profile. Consequently, the need to discover new bioactive compounds is growing, necessitating simple and rapid screening methods [1]. Traditional and folk medicine practitioners have used a variety of medicinal plants to cure various diseases. Some of these medicinally valuable phytochemicals have been consumed over generations through indigenous food habits. Phytochemical investigations of natural extracts, particularly those used in traditional food/medicine, reveal numerous compounds, and chemical screenings offer insights into the composition of their secondary metabolites.

Phenolic compounds in the human diet have been linked to potential health benefits, such as reducing the risk of chronic diseases, including diabetes, cancer, Alzheimer's, and cardiovascular diseases [2]. For instance, Tang et al. reported that high consumption of turmeric, which contains the polyphenol curcumin, may contribute to the lower incidence of Alzheimer's disease in certain populations [3]. Flavonoids, the most prevalent and widely distributed group of plant phenolics, are recognized for their antioxidant properties. Their anti-inflammatory, antioxidant, anticancer, and antimutagenic properties allow them to modulate key enzymatic activities and inhibit a range of enzymes [4]. Terpenoids, which are volatile and lipid-soluble secondary metabolites, represent a major class of plant secondary metabolites, originating from glycolytic or acetyl-CoA intermediates. Monoterpenes are widely utilized in health care and culinary applications. Diterpenes are of significant interest in medicinal research, with compounds such as ingenol-3-angelate

and paclitaxel demonstrating efficacy in cancer treatment. Triterpenes are commonly used in pharmaceutical, food, and cosmetic industries. Polyterpenes function as viscosity modifiers, such as in waxes, and serve as co-tackifiers in various industrial applications [4]. The separation of complicated mixtures into individual phytoconstituents is aided by many chromatographic procedures.

Thin-layer chromatography (TLC) is a planar chromatographic technique rooted in the early foundations of chromatography, and is known for its speed and cost-effectiveness in analyzing plant extracts and natural products globally. Its versatility across different extract polarities and capacity for simultaneous analysis of multiple samples make TLC an efficient tool for rapid screening [1]. Widely applied in food quality assessment and safety control, TLC can identify compounds influencing food stability/quality, such as antioxidants, antimicrobial, and antibrowning agents, as well as contaminants like pesticides. Additionally, TLC detects substances that may impact nutrient absorption, metabolism, and excretion, including enzyme inhibitors that could offer health benefits to consumers [5]. One benefit of TLC in this context is its ability to perform analyses without the need for complex and costly equipment.

Phenolic compounds do not fall in the category of nutrients but possess various bioactive properties and provide health-protective effects [6]. They can be examined by chromatographic techniques, but before analyzing a new plant sample, it is convenient to design an appropriate solvent system. Although TLC is a basic, adaptable, and rapid chromatographic technique, the selection of an efficient solvent system is always a tedious and time-consuming process [7]. Flavonoids/phenolic compounds have been analyzed by TLC using different solvent systems in varied ratios like chloroform–methanol: 27:0.3 [8], ethyl acetate–toluene–formic acid: 5:3.5:0.5 [9], ethyl acetate–butanol–formic acid: 2.5:1.5:0.5 [8], butanol–acetic acid–water: 4:1:5, and others.

Phenolic compounds possess different polarities based on their structure, the number and position of hydroxyl groups present, and the presence or absence of sugar moieties. Dietary flavonoids are mostly present in their glycosidic forms, but they can also exist as aglycones. This poses a challenge when it comes to choosing the appropriate solvent system for their separation and analysis by TLC. Usually, researchers have to select multiple solvent systems to achieve efficient separation of natural products belonging to different polarity levels and structural classes. Therefore, it is important to optimize the chromatographic conditions and design solvent systems that are appropriate for the efficient separation and migration of both glycosides and aglycones present in the crude plant samples. This can be achieved by optimizing the solvent system for different phenolic compounds (found in fruits and vegetables). In this study, we have compared eight different solvent systems to examine the mobilization of 18 standards of phenolic compounds.

Gas chromatography–mass spectrometry (GC–MS) is an analytical technique that is widely utilized for identifying functional groups and bioactive compounds in plant extracts. This method (also employing derivatization) is highly effective for detecting diverse chemical entities such as alkaloids, nitro compounds, long-chain hydrocarbons, organic acids, steroids, esters, and amino acids. Its efficiency is attributed to the minimal volume of plant extract required and its exceptional sensitivity, precision, and ability to separate complex mixtures [10]. In this study, the GC–MS technique was employed to identify the volatile and non-polar phytochemical constituents of *Panchkuta*, a traditional food combination made from five plants, namely *Prosopis cineraria*, *Acacia senegal*, *Capparis decidua*, *Cordia dichotoma*, and *Mangifera indica* (Figure 1). The first four are underutilized desert plants.

The objective of this study is to optimize a solvent system for analyzing phenolic compounds via TLC and to perform phytochemical profiling of five plants of *Panchkuta* by TLC and GC-MS analysis.

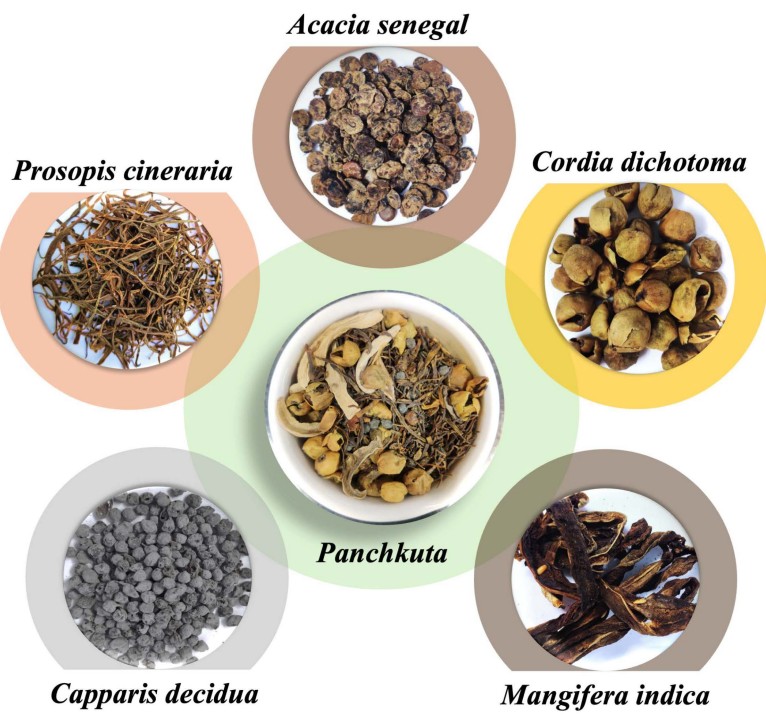

**Figure 1.** Five plants included in the traditional food combination—*Panchkuta*.

## 2. Materials and Methods

### 2.1. Reagents and Chemicals

Methanol, hexane, Polyethylene glycol (PEG) 4000, 2-aminoethyl diphenylborinate (AEPB), the standards rutin hydrate, quercetin dihydrate, quercetin-β-D-glucoside, daidzin, daidzein, genistin, kaempferol, naringin, naringenin, esculin hydrate, hydroquinone, puerarin, biochanin, and hesperetin were purchased from Sigma-Aldrich (St. Louis, MO, USA), while myricetin, apigenin, resveratrol, and scopoletin were procured from TCI (Shanghai, China). The TLC plates (precoated with silica gel 60 $F_{254}$) were purchased from Merck (Darmstadt, Germany).

### 2.2. Sample Collection and Extraction

Commercially available dried pods of *P. cineraria* (PC; local name: Sangri), berries of *C. decidua* (CD; local name: Ker), seeds of *A. senegal* (AS; local name: Kumatiya), the fruit of *C. dichotoma* (CDI; local name: Lasora), and pulp of *M. indica* (MI; Mango) were purchased from a local grocery store (Nagaur district, Rajasthan, India; N 27°11′55.0644″, E 73°44′4.9848″).

A sample of *Panchkuta* was procured from Rajasthan, India (26.2389° N, 73.0243° E). Each ingredient was segregated and weighed to ascertain the specific ratios employed in the regional mixing practices. *P. cineraria*, *A. senegal*, *C. decidua*, *C. dichotoma*, and *M. indica* were present in different ratios (by weight): specifically, 14:13:8:4:1.

Preparation of extracts—The dried plant samples were powdered using a mixer grinder and extracted with hexane in a 1:5 (*w/v*) ratio at room temperature for an hour in an orbital shaker incubator. This was followed by centrifugation at $3000 \times g$ for 10 min. The supernatant was filtered using Whatman filter paper (No. 1) and the pellet was extracted two more times. The hydrophobic/non-polar compounds were separated in the hexane extract and the pellet was dried at room temperature. This dried pellet was extracted thrice with 5 volumes of 80% methanol by the same process mentioned above. All the supernatants were pooled and concentrated to dryness using a rotary evaporator (Aditya Scientific, Hyderabad, India). The concentrated extracts were stored at 4 °C for further analysis.

### 2.3. Optimization of Solvent System for TLC

In this study, TLC was employed to identify phenolic compounds from partially purified methanolic plant extracts. This process involved optimizing solvent systems to enable accurate detection of reference standards and effective separation of compounds within the extracts. To achieve this, various solvent systems were employed to analyze potential phenolic compounds and their glycosidic forms. Due to the polarity differences between aglycones and glycosides, standardizing a suitable solvent system for TLC separation presented challenges, as a single system often fails to facilitate the migration of all spotted compounds on the TLC plate.

All the standards were dissolved in 100% methanol at a concentration of 1 mg/mL. After that, 10 µL of each sample was applied to the TLC plate using a micropipette and analysis was performed using 8 different solvent systems (Table 1). The plates were developed in a cylindrical glass chamber (10 cm × 11 cm) and dried in an oven at 40 °C after development. Visualization was performed 1) under UV light at 254 nm and 366 nm using CAMAG TLC visualizer (CAMAG, Muttenz, Switzerland) and 2) after the sample was sprayed with Natural Product Reagent (NPR), which is 1% methanolic AEPB (2-aminoethyl diphenylborinate) and 5% ethanolic Polyethylene glycol 4000, followed by visualization under UV (365 nm).

**Table 1.** Solvent systems employed for thin-layer chromatography.

| Solvent Systems | Solvent | Ratio |
| --- | --- | --- |
| A | Butanol–acetic acid–water | 7:2:2 |
| B | Chloroform–ethyl acetate–acetone | 5:1:4 |
| C | Chloroform–methanol | 9:1 |
| D | Chloroform–methanol–water | 26:14:4 |
| E | Ethyl acetate–formic acid–glacial acetic acid–water | 10:1.1:1.1:2.6 |
| F | Ethyl acetate–methanol–water | 15:3:2 |
| G | Ethyl acetate–toluene–formic acid | 4:4:1 |
| H | Toluene–ethyl acetate–acetone–formic acid | 20:4:2:1 |

The migration of active compounds is expressed by the retention factor ($R_f$).

$$R_f = \frac{Distance\ travelled\ by\ the\ solute}{Distance\ travelled\ by\ the\ solvent}$$

The $R_f$ values of many phytochemicals provide important information on their identity, polarity and the selection of the solvent used for their separation.

### 2.4. TLC Fingerprint Profiling of Methanolic Plant Extracts

All the five plant extracts were dissolved in 80% methanol (1 mg/mL), followed by filtration through 0.45 µm filters. A small quantity (5–10 µg) of the extract was applied to the TLC plate and analyzed using various solvent systems, as per the method discussed above. Distinct fluorescent colors indicative of specific phenolic phytochemicals appeared either instantly or within 15 min when observed under 365 nm.

### 2.5. Phytochemical Profiling of the Panchkuta Hexane Extract by GC-MS Analysis

The hexane extract of crude *Panchkuta* (PC: AS: CD: CDI: MI: 14:13:8:4:1) was characterized by GC-MS analysis using Shimadzu Scientific (Kyoto, Japan) GC-MS TQ8050 NCI equipment in pulsed splitless mode. The samples were separated on a Rxi-5Sil MS column (Restek, Bellefonte, PA, U.S.A) with dimensions of 30 mm × 0.25 mm (ID) and a 0.25 µm film thickness. Helium was used as a carrier gas with a flow rate of 1 mL/min. The following parameters were used: injector temperature—280 °C; detector temperature—305 °C; column head pressure maintained at 13 psi; interface temperature—280 °C; and ionization

energy—70 eV. Initially, the column temperature was maintained at 80 °C for 1 min, then set to 220 °C at a rate of 10 °C/min, 220–310 °C at a rate of 20 °C/min with a final 6 min of hold time. The sample injection volume was 2 μL, and mass spectra were scanned from 50 to 700 $m/z$ at a rate of 1.5 scans/s [11]. The fragmentation patterns of the mass spectra were compared to the NIST library for identification.

## 3. Results and Discussion

### 3.1. Optimization of Solvent System for TLC of Phenolic Phytochemicals

TLC remains a rapid and cost-effective method for separating various classes of natural products extracted from plants. Although more advanced separation techniques (like LC-MS) have been developed in recent years, TLC is still widely used for rapid and preliminary identification of compounds. Initial standardization of the TLC method was performed using commercially available standards for phenolics/flavonoids.

The $R_f$ values of standards in the chosen solvent systems are summarized in Table 2.

**Table 2.** $R_f$ values of phenolic phytochemicals in the chosen solvent systems of TLC. Refer to Table 1 for solvent system composition.

| S. No. | Phenolic Phytochemical | $R_f$ Values in Respective Solvent Systems | | | | | | | |
|---|---|---|---|---|---|---|---|---|---|
| | | **A** | **B** | **C** | **D** | **E** | **F** | **G** | **H** |
| 1 | Rutin hydrate | 0.46 | 0 | 0 | 1 | 0.63 | 0.72 | 0 | 0 |
| 2 | Quercetin dihydrate | 1 | 0.64 | 0.63 | 1 | 0.96 | 0.98 | 0.66 | 0.32 |
| 3 | Daidzin | 0.63 | 0.09 | 0.2 | 1 | 0.68 | 0.80 | 0.2 | 0 |
| 4 | Genistin | 0.7 | 0.11 | 0.23 | 1 | 0.71 | 0.83 | 0.29 | 0 |
| 5 | Kaempferol | 1 | 0.83 | 0.8 | 1 | 0.96 | 1 | 0.70 | 0.45 |
| 6 | Myricetin | 1 | 0.5 | 0.41 | 0.96 | 0.93 | 0.93 | 0.58 | 0.16 |
| 7 | Apigenin | 1 | 0.87 | 0.9 | 1 | 0.96 | 1 | 0.66 | 0.40 |
| 8 | Naringin | 0.66 | 0.32 | 0.3 | 0.96 | 0.68 | 0.77 | 0.29 | 0 |
| 9 | Scopoletin | 1 | 0.87 | 0.96 | 0.96 | 0.91 | 0.90 | 0.60 | 0.35 |
| 10 | Resveratrol | 1 | 0.72 | 0.53 | 1 | 1 | 0.91 | 0.60 | 0.37 |
| 11 | Esculin hydrate | 0.5 | 0.08 | 0.16 | 0.84 | 0.77 | 0.58 | 0.12 | 0 |
| 12 | Daidzein | 1 | 0.74 | 0.75 | 1 | 1 | 0.91 | 0.56 | 0.31 |
| 13 | Hydroquinone | 1 | 0.79 | 0.71 | 0.98 | 1 | 0.91 | 0.59 | 0.42 |
| 14 | Naringenin | 1 | 0.82 | 0.88 | 1 | 1 | 0.93 | 0.67 | 0.47 |
| 15 | Quercetin-β-D-glucoside | 0.65 | 0 | 0 | 0.78 | 0.80 | 0.7 | 0.19 | 0 |
| 16 | Puerarin | 0.65 | 0.18 | 0 | 0.82 | 0.82 | 0.63 | 0.24 | 0 |
| 17 | Biochanin | 1 | 0.89 | 1 | 1 | 1 | 0.96 | 0.70 | 0.63 |
| 18 | Hesperetin | 1 | 0.86 | 1 | 1 | 1 | 0.98 | 0.67 | 0.50 |

Out of the eight solvent systems tested, it was observed that the solvent system A (butanol–acetic acid–water: 7:2:2) and, to a lesser extent, solvent system E (ethyl acetate–formic acid–glacial acetic acid–water: 10:1.1:1.1:2.6), showed the proper separation of glycosides, which are relatively polar compounds (rutin hydrate, daidzin, genistin, naringin, esculin hydrate, quercetin-β-D-glucoside, and puerarin), but the aglycones, which are relatively less polar compounds (with $R_f$ value of 1), moved along with the solvent front because of the higher polarity of the mobile phase. Solvent systems H (toluene–ethyl acetate–acetone–formic acid: 20:4:2:1) and G (ethyl acetate–toluene–formic acid: 4:4:1), showed the proper separation of the aglycones (quercetin dihydrate, kaempferol, apigenin, scopoletin, resveratrol, daidzein, hydroquinone, naringenin, puerarin, and hesperetin), but in these solvent systems, the glycosides did not migrate from the origin (Figure 2).

Myricetin showed the proper band only in solvent system G. Although the migration of non-polar compounds was also achieved using the solvent systems B and C, the resolution of bands was poor, with smear formation and tailing of compounds.

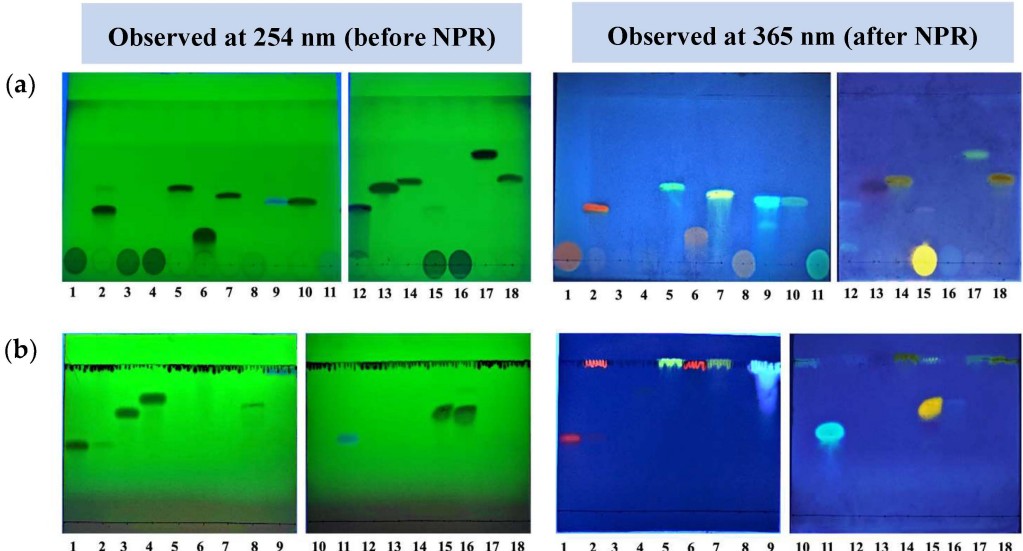

**Figure 2.** TLC images of phenolic compounds observed at 254 nm and 366 nm. The solvent system used is (**a**) toluene–ethyl acetate–acetone–formic acid: 20:4:2:1, and (**b**) butanol–acetic acid–water: 7:2:2. The colors of the compounds and their respective classes are given in Table 3. Samples: 1—rutin hydrate, 2—quercetin dihydrate, 3—daidzin, 4—genistin, 5—kaempferol, 6—myricetin, 7—apigenin, 8—naringin, 9—scopoletin, 10—resveratrol, 11—esculin hydrate, 12—daidzein, 13—hydroquinone, 14—naringenin, 15—quercetin-β-D-glucoside, 16—puerarin, 17—biochanin, 18—hesperetin.

**Table 3.** Class and color of compounds before and after spraying NPR on silica gel F254 TLC plate.

| Class of Phenolic Compounds | Phenolic Phytochemical | Color in UV Without NPR (254 nm) | Color in UV with NPR (365 nm) |
|---|---|---|---|
| Flavonols and their glycosides | Rutin hydrate | Dark mauve | Orange |
| | Quercetin dihydrate | Dark mauve | Orange-yellow |
| | Quercetin-β-D-glucoside | Faint Orange | Yellow/orange |
| | Kaempferol | Dark mauve | Yellow-green |
| | Myricetin | Dark mauve | Dark Orange |
| Isoflavones and their glycosides | Daidzin | Mauve | Faint orange |
| | Genistin | Mauve | Faint orange |
| | Daidzein | Mauve | Faint blue |
| | Puerarin | Mauve | Faint blue |
| | Biochanin | Dark mauve | Faint green |
| Flavones and their glycosides | Apigenin | Dark mauve | Yellow-green |
| Flavanone and their glycosides | Naringin | Pale yellow | Yellow-brown |
| | Naringenin | Faint mauve | Faint brown |
| | Hesperetin | Mauve | Brown |
| Coumarins | Scopoletin | Light blue | Bright blue |
| | Esculin hydrate | Blue | Bright blue |

**Table 3.** *Cont.*

| Class of Phenolic Compounds | Phenolic Phytochemical | Color in UV Without NPR (254 nm) | Color in UV with NPR (365 nm) |
|---|---|---|---|
| Stilbene | Resveratrol | Violet blue | Blue |
| Phenol | Hydroquinone | Dark orange | Faint purple |

The phenolic compounds showed characteristic colors in both 254 nm and 365 nm, before and after spraying NPR. Table 3 summarizes the color properties of different compounds. The indicative classes/categories of phenolic compounds to which these individual compounds belong (column 1 of Table 3) were deduced by referring to the TLC atlas by Sabine Bladt (2009) [12]. The respective $R_f$ values have already been mentioned in Table 2.

*3.2. TLC Fingerprint Profile of Five Plants of Panchkuta*

The chemical diversity of plant compounds, ranging from highly non-polar to highly polar, necessitates selective extraction approaches to effectively isolate metabolites. Generally, the extraction process begins with a non-polar solvent and then proceeds with solvents of progressively higher polarity. This stepwise method enables the sequential separation of compounds by polarity, allowing targeted compounds to be fractionated in a specific solvent type. Initially, hydrophobic or non-polar compounds are extracted using solvents like hexane. More polar compounds are subsequently extracted with higher-polarity solvents, such as methanol or ethanol.

When the methanolic plant extracts were loaded onto the TLC plates, it was observed that a single solvent system was not capable of separating all the constituents (flavonoids and their glycosides) present in the extracts. As discussed in Section 3.1, solvent systems A (butanol–acetic acid–water: 7:2:2) and H (toluene–ethyl acetate–acetone–formic acid: 20:4:2:1) were found best to separate polar and non-polar compounds, respectively. Hence, for the phytochemical fingerprinting of the methanolic plant extracts, we chose these two solvent systems out of the eight solvent systems.

The TLC images from solvent system H revealed that most of the sample remained concentrated at the origin, indicating insufficient separation during the run (Figure 3a). This suggests that solvent system H lacked the necessary polarity to effectively mobilize the compounds in the methanolic plant extracts. The retention of compounds at the origin also suggests that these compounds are highly polar, possibly due to conjugation with sugar molecules, which increases their polarity and limits their movement. To address this, the extracts were subjected to TLC with solvent system A, consisting of butan-1-ol, acetic acid, and water (BAW) in a 7:2:2 ratio, used as the mobile phase.

Solvent system A is a more polar mobile phase, which facilitates migration of the relatively more polar compounds in the extracts. As shown in Figure 3b, the majority of the compounds were able to migrate on the TLC plate with this solvent system, suggesting that they may be in conjugated forms. The color variations in the bands on the TLC plates indicated the presence of multiple phenolics, flavonoids, and isoflavonoids in the defatted methanolic extracts. The unique band patterns among the samples resulted in a distinct fingerprint for the UV-active phytochemicals (mainly phenolics) within each sample. To further improve compound separation, we slightly reduced the polarity of solvent system A by adjusting the solvent ratios from BAW—7:2:2 to BAW—12:1.5:1.5, forming a modified solvent system. This modification resulted in better separation, as reflected in new, differently colored bands across all plant samples under both 254 nm and 365 nm UV light (Figure 3c).

Under 254 nm UV light, phenolic compounds appeared as dark spots on a green background due to their ability to absorb UV light and quench the plate's green fluorescent indicator. In contrast, under 365 nm UV light, different-colored fluorescent zones were observed, as certain compounds exhibited fluorescence when excited by longer-wavelength UV light. The orange-red-, blue-, purple-, yellow-, and faint green-colored

bands indicate the presence of different classes of phenolic/flavonoid compounds, including flavonols, isoflavonoids, phenolic acids, and anthocyanins. Among these, quercetin is the most studied flavonol found in edible plants, with onion being the richest source, while tea and grapes contain lower amounts. It provides protection to cellular structures and blood vessels from oxidative stress, exhibiting both antioxidant and anti-inflammatory properties [13]. Myricetin, another flavonol, has demonstrated health benefits, including antidiabetic, hepatoprotective, neuroprotective, cardioprotective, antimicrobial, analgesic, and antihypertensive effects in both in vitro and in vivo studies [14]. Isoflavones, primarily found in legumes, also display antioxidant, anti-angiogenic, estrogenic, osteoclastic, and immunosuppressive activities [15]. The identity of the compounds was also validated further through LC-MS of these plants, individually as well as in combination in the form of *Panchkuta,* which we have previously reported [16,17].

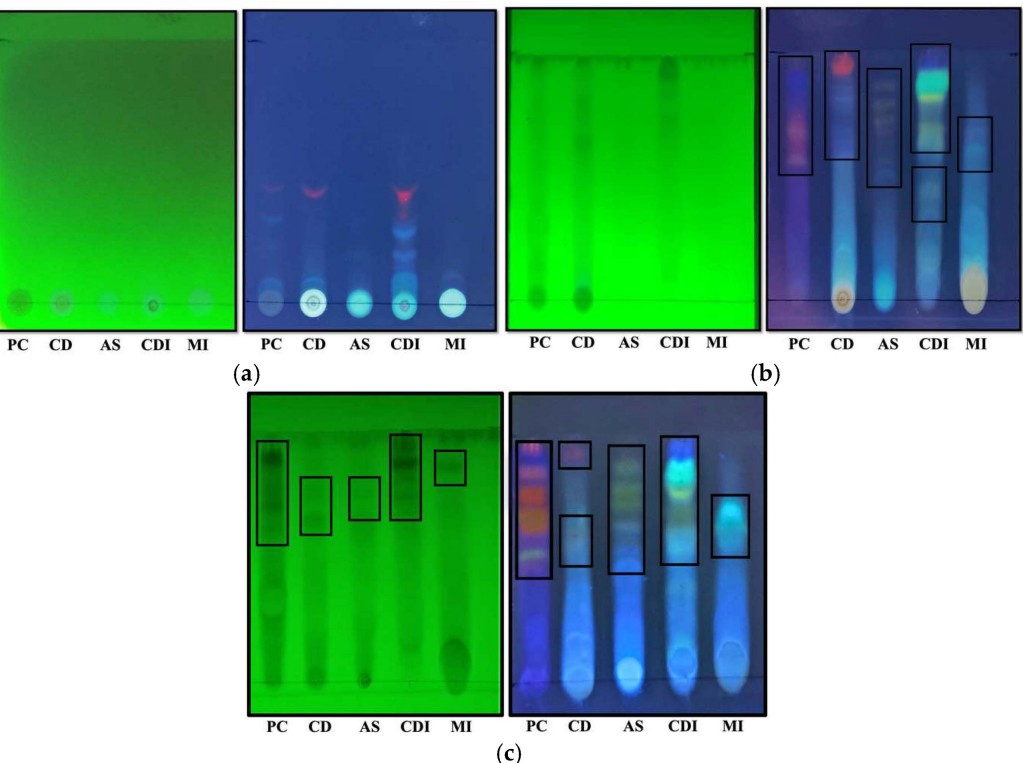

**Figure 3.** Phytochemical fingerprinting of five plants of *Panchkuta* by TLC, visualized under 254 nm and 365 nm. The solvent systems used are (**a**) toluene–ethyl acetate–acetone–formic acid: 20:4:2:1 (solvent system H), (**b**) butanol–acetic acid–water: 7:2:2 (solvent system A), and (**c**) butanol–acetic acid–water: 12:1.5:1.5 (modified solvent system A). The boxes indicate the presence of major metabolites belonging to different classes of phenolic compounds (mentioned in Table 3). PC—*P. cineraria*, CD—*C. decidua*, AS—*A. senegal*, CDI—*C. dichotoma*, and MI—*M. indica*.

### 3.3. Phytochemical Profile of Panchkuta Hexane Extract by GC-MS Analysis

The byproducts of phenolic purification (volatile non-polar metabolites), such as terpenoids, sterols, hydrocarbons, and fatty acids, are often discarded in hexane-based plant extract purification processes despite their significant health benefits. These compounds, while not the primary target of phenolic extraction, have diverse biological activities and therapeutic potential. For the same purpose, GC-MS analysis of the hexane extract was performed to identify the non-polar volatile natural metabolites present in *Panchkuta*. Figure 4 shows the GC-MS chromatogram, and the identified compounds (their mass spectra were compared with those of reference compounds available in the NIST library) belonging to different classes of phytochemicals are shown in Table 4. The results showed the presence of different classes of compounds, including terpenes, hydrocarbons (alkane and aromatic),

fatty acids, vitamins, and phytosterols. The major bioactive metabolites present in the hexane extract were gamma-sitosterol (14.05%), dotriacontane (8.06%), stigmasterol (6.70%), campesterol (5.61%), 1-hexacosanol (5.25%), oleic acid (4.83%), palmitic acid (3.48%), and others. Some of these compounds have been reported to possess various bioactivities; for example, squalene has immunity enhancement, hypolipidemic, antioxidant, and anti-tumor properties [18], oleic acid has antibacterial activity [19], stigmasterol shows anticancer, antifungal, and anti-inflammatory activities, and tocopherols (a form of vitamin E) have antioxidant potential [20]. p-cymene, a monoterpene, is used as a flavoring agent and has been reported to possess antioxidant, anti-inflammatory, antidiabetic, anti-tumor, and analgesic properties [21]. Alkane hydrocarbons such as dodecane, eicosane, and heneicosane have been reported to show anti-bacterial properties [22]. Phytol, a diterpene alcohol, has anti-inflammatory [23] and anti-diarrheal [24] properties.

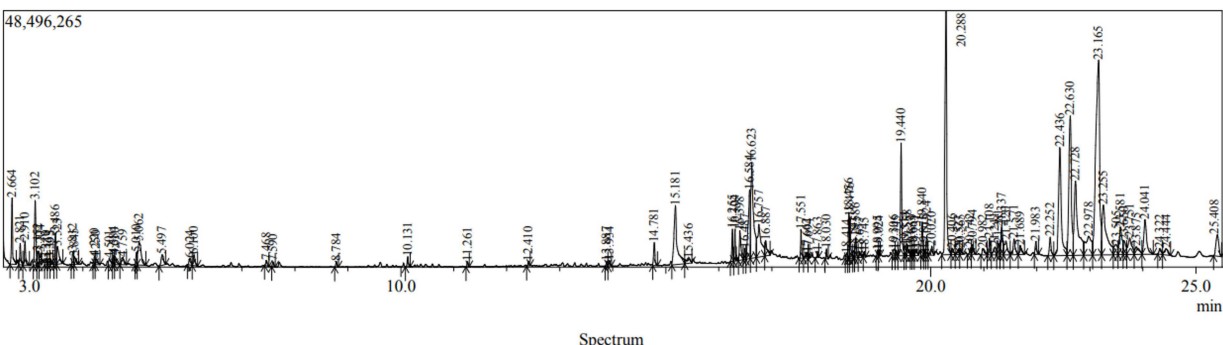

**Figure 4.** GC-MS chromatogram of *Panchkuta* hexane extract.

**Table 4.** Phytochemical profile of hexane extract of *Panchkuta* analyzed by GC-MS.

| S. No. | Rt (min) | Area (%) | Indicated Metabolite | Type of Compound |
|--------|----------|----------|----------------------|------------------|
| 1. | 2.664 | 1.41 | 2-ethyl-3-methylbutanal | Aliphatic aldehyde |
| 2. | 2.910 | 0.53 | Butanoic acid, 3-oxo-, 1-methylpropyl ester | Carboxylic acid ester |
| 3. | 3.102 | 1.60 | 3-hexen-2-one | Methyl ketone |
| 4. | 4.501 | 0.27 | p-cymene | Monoterpene |
| 5. | 5.062 | 1.39 | 1,2,3,5-tetramethyl-benzene | Aromatic hydrocarbon |
| 6. | 6.100 | 0.36 | Dodecane | n-alkane hydrocarbon |
| 7. | 8.784 | 0.12 | Tetradecane | Alkane hydrocarbon |
| 8. | 11.261 | 0.11 | Heneicosane | Alkane hydrocarbon |
| 9. | 12.410 | 0.10 | Eicosane | Alkane |
| 10. | 14.781 | 0.56 | Hexadecanoic acid, methyl ester | Fatty acid methyl ester |
| 11. | 15.181 | 3.48 | n-Hexadecanoic acid (palmitic acid) | Fatty acid |
| 12. | 16.304 | 0.62 | 9-Octadecenoic acid, methyl ester | Fatty acid methyl ester |
| 13. | 16.398 | 0.94 | Phytol | Diterpene alcohol |
| 14. | 16.487 | 0.24 | Methyl stearate | Fatty acid methyl ester |
| 15. | 16.584 | 2.77 | 9,12-Octadecadienoic acid (linoleic acid) | Fatty acid |
| 16. | 16.623 | 4.83 | Oleic acid | Fatty acid |
| 17. | 16.757 | 2.32 | Octadecanoic acid (stearic acid) | Fatty acid |
| 18. | 18.456 | 0.83 | Methyl 5,11,14-eicosatrienoate | Fatty acid methyl ester |

**Table 4.** *Cont.*

| S. No. | Rt (min) | Area (%) | Indicated Metabolite | Type of Compound |
|--------|----------|----------|----------------------|------------------|
| 19. | 18.476 | 0.88 | Glycidyl oleate | Carboxylic ester and epoxide |
| 20. | 18.586 | 0.45 | Pentatriacontane | Alkane hydrocarbon |
| 21. | 18.745 | 0.10 | Phthalic acid, di(6-methylhept-2-yl) ester | Aromatic carboxylic acid ester |
| 22. | 19.440 | 2.58 | Tetrapentacontane | Alkane hydrocarbon |
| 23. | 19.558 | 0.40 | 13-Methylheptacosane | Alkane hydrocarbon |
| 24. | 19.633 | 0.11 | 1,4-benzenedicarboxylic acid, bis(2-ethylhexyl) ester | Carboxylic acid ester |
| 25. | 19.924 | 0.37 | Squalene | Triterpenoid hydrocarbon |
| 26. | 19.731 | 0.24 | 2-Methyloctacosane | Alkane hydrocarbon |
| 27. | 20.288 | 8.06 | Dotriacontane | Alkane hydrocarbon |
| 28. | 21.108 | 0.44 | gamma-tocopherol | Vitamin |
| 29. | 21.337 | 0.96 | Triacontanol | Fatty alcohol |
| 30. | 21.420 | 0.76 | 2-(2,2-Diphenylacetamido)thiazole | Phenyl thiazole |
| 31. | 21.571 | 0.66 | alpha-tocopherol-beta-D-mannoside | Vitamin |
| 32. | 22.436 | 5.61 | Campesterol | Phytosterol |
| 33. | 22.630 | 6.70 | Stigmasterol | Phytosterol |
| 34. | 22.728 | 5.25 | 1-Hexacosanol | Fatty alcohol |
| 35. | 22.978 | 2.13 | 26,27-Dinorergosta-5,24-dien-3-ol, (3 beta)- | Phytosterol |
| 36. | 23.165 | 14.05 | gamma-Sitosterol | Phytosterol |
| 37. | 23.255 | 4.13 | Stigmasta-5,24(28)-dien-3-ol, (3.beta,24Z) | Phytosterol |
| 38. | 23.751 | 1.17 | Lup-20(29)-en-3-one | Triterpene |
| 39. | 24.041 | 1.92 | Lup-20(29)-en-3-ol, acetate, (3 beta)- | Triterpene |
| 40. | 24.322 | 0.23 | gamma-Sitostenone | Phytosterol |

Other researchers have also reported the presence of similar phytoconstituents in the five plants of *Panchkuta*, including squalene, eicosane, dotriacontane, and hexadecenoic acid in *P. cineraria* [25]; hexadecanoic acid-methyl ester, methyl stearate, 9,12-octa-decadienoic acid [26], phytol, stigmasterol, squalene, oleic acid, and tocopherol [27] in *C. decidua*; hexade-canoic acid and 9,12-octa-decadienoic acid in *A. senegal*; phytol, hexa-decanoic acid-methyl ester, n-hexadecanoic acid, eicosane [28], heneicosane, squalene, tetradecane, and dotria-contane in *C. dichotoma*; p-cymene and heneicosane [29,30] in *M. indica*. Integrating these products into nutraceuticals, cosmetics, or pharmaceuticals could unlock their potential, promoting holistic utilization of plant resources.

## 4. Conclusions

This study reveals the potential of the TLC technique used for the analysis of phyto-chemicals in plant extracts. Based on the current results, solvent systems A (butanol–acetic acid–water: 7:2:2) and H (toluene–ethyl acetate–acetone–formic acid: 20:4:2:1) were found to be more efficient for the migration of polar and non-polar phenolic compounds, respectively. However, solvent H did not facilitate the movement of diverse range of compounds in the selected five plants due to its relatively high non-polar nature. Modified solvent A (BAW-12:1.5:1.5) facilitated the movement of the majority of metabolites present in the methanolic plant extracts. The results of the current work will help researchers to save time by provid-ing efficient solvent systems for the separation of polar and non-polar phenolic compounds. The TLC of the five plants (*P. cineraria*, *C. decidua*, *C. dichotoma*, *A. senegal*, and *M. indica*) revealed a diverse phytochemical fingerprint and showed the presence of various classes

of phenolic compounds. The appearance of orange, blue, green, yellow, brown, and purple fluorescent bands were indicative of flavonols, isoflavones, coumarins, flavones, flavanones, and phenols, respectively. The GC-MS analysis of *Panchkuta* hexane extract showed the presence of volatile non-polar bioactive metabolites (linoleic acid, oleic acid, stearic acid, squalene, gamma-tocopherol, campesterol, stigmasterol, and gamma-sitosterol). Chromatographic analysis of traditional plants/food combinations like *Panchkuta*, as reported in the current article, helps to identify various bioactive metabolites possessing health-benefitting properties (anticancer, antioxidant, anti-inflammatory, and antifungal properties, among others). Fruits/pods derived from such underutilized desert plants could serve as a natural source of sustainable food and can contribute to addressing the United Nation's Sustainable Development Goals (SDGs), like SDG 2 (end hunger, achieve food security and improved nutrition, and promote sustainable agriculture) and SDG 3 (ensure healthy living and promote well-being).

**Author Contributions:** T.J.: Writing—original draft, Methodology, Investigation, Formal analysis, Data curation. P.R.D.: Writing—review and editing, Validation, Supervision, Conceptualization. P.K.S.: Writing—review and editing, Visualization, Validation, Supervision, Conceptualization. All authors have read and agreed to the published version of the manuscript.

**Funding:** This research received no external funding.

**Institutional Review Board Statement:** Not applicable.

**Informed Consent Statement:** Not applicable.

**Data Availability Statement:** No data were used for the research described in the article.

**Acknowledgments:** The authors are grateful to the administration of Birla Institute of Technology and Science (BITS), Pilani, Pilani Campus, for infrastructural and logistic support. Tripti Joshi is grateful to the University Grants Commission, New Delhi, India, for providing the Senior Research Fellowship. The authors are thankful to Sophisticated Instrumentation Facility, BITS Pilani—Pilani Campus (India) for assisting with the GC–MS instrumentation.

**Conflicts of Interest:** The authors declare no conflicts of interest.

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
