# Peer review of "Chromatographic Analysis of Bioactive Metabolites from a Traditional Food Combination of (Semi) Arid Regions—Panchkuta: Insights for Sustainable Functional Foods Development"

_blsf, doi:10.3390/blsf38010005_

Round 1

Reviewer 1 Report

Comments and Suggestions for Authors

The study highlights the significance of plants as valuable resources for high-value products across industries like pharmaceuticals, cosmetics, and food. The paper focuses on thin-layer chromatography (TLC) and gas chromatography-mass spectrometry (GC-MS) as pivotal techniques for analyzing plant extracts and identifying bioactive compounds. The study compares eight solvent systems to optimize separating 18 phenolic standards. The analysis underscores the need for using tailored solvent systems to efficiently separate compounds of varying polarities and structures. The study bridges the gap between traditional uses of plants and modern analytical techniques, emphasizing their complementary roles in discovering new bioactive compounds.

Comments:

The GC-MS analysis focuses on volatile and non-polar compounds, excluding polar compounds, which may provide additional insights into the phytochemical profile of Panchkuta. Do the authors have a comment about it?

The study briefly mentions the medicinal and nutritional value of the plants but does not elaborate on how the identified compounds could translate to practical applications. The authors could briefly include this in the introduction.

In the discussion, the authors could link the identified compounds to specific bioactivities or traditional uses, strengthening the study's relevance to medicinal and nutritional applications.

Author Response

Response to Reviewers

Thank you very much for taking the time to review this manuscript. Please find the detailed responses below and the corresponding revisions highlighted in the revised manuscript.

Comments and Suggestions for Authors

The study highlights the significance of plants as valuable resources for high-value products across industries like pharmaceuticals, cosmetics, and food. The paper focuses on thin-layer chromatography (TLC) and gas chromatography-mass spectrometry (GC-MS) as pivotal techniques for analyzing plant extracts and identifying bioactive compounds. The study compares eight solvent systems to optimize separating 18 phenolic standards. The analysis underscores the need for using tailored solvent systems to efficiently separate compounds of varying polarities and structures. The study bridges the gap between traditional uses of plants and modern analytical techniques, emphasizing their complementary roles in discovering new bioactive compounds.

Comment 1: The GC-MS analysis focuses on volatile and non-polar compounds, excluding polar compounds, which may provide additional insights into the phytochemical profile of Panchkuta. Do the authors have a comment about it?

Response 1: Thank you for pointing this out. We agree that the inclusion of polar compounds provide additional insights into the phytochemical profile of Panchkuta. As mentioned on page 9, section 3.2, paragraph 5, and line 267-269, we have already reported the identification of polar phenolic compounds (by LC-MS analysis) present in the five plants of Panchkuta. Kindly find the articles for your reference.

Joshi, T.; Agrawal, K.; Mangal, M.; Deepa, P.R.; Sharma, P.K. Measurement of Antioxidant Synergy between Phenolic Bioactives in Traditional Food Combinations (Legume/Non-Legume/Fruit) of (Semi) Arid Regions: Insights into the Development of Sustainable Functional Foods. Discov. Food 2024, 4, 11, doi:10.1007/s44187-024-00082-y.

Joshi, T.; Puri, S.; Deepa, P.R.; Sharma, P.K. Bioactivity-Guided Purification and Characterization of Antioxidant, Anti-Gout and Anti-Diabetic Polyphenols from Panchkuta: A Traditional Food Combination of (Semi) Arid Regions. Food Chem. Adv. 2024, 5, 100839, doi:10.1016/j.focha.2024.100839.

Comment 2: The study briefly mentions the medicinal and nutritional value of the plants but does not elaborate on how the identified compounds could translate to practical applications. The authors could briefly include this in the introduction.

Response 2: Thank you for the suggestion. We have now included a paragraph on the bioactivities of different classes of compounds present in the Panchkuta sample. Kindly refer to page 1, section 1 (introduction), paragraph 2, and lines 32-45.

Comment 3: In the discussion, the authors could link the identified compounds to specific bioactivities or traditional uses, strengthening the study's relevance to medicinal and nutritional applications.

Response 3: Thank you for your suggestion. We have now incorporated the bioactivities of various classes of phenolic compounds identified through TLC. Although, we had discussed the bioactivities of certain compounds identified via GC-MS analysis, we have now included the pharmacological potential of a few more compounds. Kindly refer to page 9, section 3.2, paragraph 5, and lines 260-267; page 10, section 3.3, paragraph 1, and lines 288-292.
